

# Evaluation of Four Ground-based Retrievals of Cloud Droplet Number Concentration in Marine Stratocumulus with Aircraft *In Situ* Measurements

Damao Zhang[1], Andrew M. Vogelmann[2], Fan Yang[2], Edward Luke[2], Pavlos Kollias[2,3], Zhien Wang[4], Peng Wu[1], William I. Gustafson Jr.[1], Fan Mei[1], Susanne Glienke[1], Jason Tomlinson[1], and Neel Desai[5]

[1]Pacific Northwest National Laboratory, Richland, Washington, USA
[2]Brookhaven National Laboratory, Upton, New York, USA
[3]School of Marine and Atmospheric Sciences, Stony Brook University, New York, USA
[4]College of Arts and Sciences, University of Colorado Boulder, CO, USA
[5]Department of Meteorology and Climate Science, San Jose State University, CA, USA

*Correspondence to*: Damao Zhang (damao.zhang@pnnl.gov)

**Abstract.** Cloud droplet number concentration ($N_d$) is crucial for understanding aerosol-cloud interactions (ACI) and associated radiative effects. We present evaluations of four ground-based $N_d$ retrievals based on comprehensive datasets from the Atmospheric Radiation Measurements (ARM) Aerosol and Cloud Experiments in the Eastern North Atlantic (ACE-ENA) field campaign. The $N_d$ retrieval methods use ARM ENA observatory ground-based remote sensing observations from a Micropulse lidar, Raman lidar, cloud radar, and the ARM NDROP Value-added Product (VAP), all of which also retrieve cloud effective radius ($r_e$). The retrievals are compared against aircraft measurements from the Fast-Cloud Droplet Probe (FCDP) and the Cloud and Aerosol Spectrometer (CAS) obtained from low-level marine boundary layer clouds on 12 flight days during summer and winter seasons. Additionally, the *in situ* measurements are used to validate the assumptions and characterizations used in the retrieval algorithms. Statistical comparisons of the probability distribution function (PDF) of the $N_d$ and cloud $r_e$ retrievals with aircraft measurements demonstrate that these retrievals align well with *in situ* measurements for overcast clouds, but they may substantially differ for broken clouds or clouds with low liquid water path (LWP). The retrievals are applied to four years of ground-based remote sensing measurements of overcast marine boundary layer clouds at the ARM ENA observatory to find that $N_d$ ($r_e$) values exhibit seasonal variations, with higher (lower) values during the summer season and lower (higher) values during the winter season. The ensemble of various retrievals using different measurements and retrieval algorithms such as those in this paper can help to quantify $N_d$ retrieval uncertainties and identify reliable $N_d$ retrieval scenarios. Of the retrieval methods, we recommend using the using the Micropulse lidar-based method given its good agreement with *in situ* measurements, it has less sensitivity to issues arising from precipitation and low cloud LWP/optical depth, and it has broad applicability by functioning for both day and nighttime conditions.



## 1 Introduction

Clouds play a crucial role in regulating the energy balance and water cycle of the Earth (Stephens et al., 2012). By reflecting incoming solar radiation back to space (the 'albedo effect') and trapping outgoing longwave radiation (the 'greenhouse effect'), they cause both cooling and warming effects on Earth's climate. On a global scale, clouds have a net cooling effect of approximately 20 $Wm^{-2}$, which is more than five times greater than the warming effect caused by doubling the concentration of atmospheric $CO_2$ (IPCC 2021). Hence, even small changes in cloud properties, such as those induced by anthropogenic activities like aerosol emissions, can significantly impact Earth's climate sensitivity (Zelinka et al., 2017). Aerosols indirectly affect cloud properties by serving as cloud condensation nuclei (CCN) or ice nucleation particles. Such effects can increase the concentration of cloud droplets ($N_d$) and decrease their sizes, which can substantially alter cloud radiative properties and precipitation efficiency (Twomey 1977; Albrecht 1989). Recent studies have also revealed that aerosol-cloud interactions (ACI) are strongly influenced by atmospheric dynamics and thermodynamic conditions, as well as the physical properties and chemical compositions of aerosols (Chen et al., 2016; Fan et al., 2016). The uncertainty in the magnitude of ACI remains the largest source of uncertainty in estimates of climate forcing (IPCC 2021; Regayre et al., 2014). $N_d$, which is a direct link between cloud properties and aerosol concentrations, is of utmost importance in improving our understanding of ACI processes and quantifying their effective radiative forcing (Rosenfeld et al., 2019).

To improve the representation of clouds in weather and climate models, it is essential to validate modeled $N_d$ against observations (Storelvmo et al., 2006; Moore et al., 2013; Gryspeerdt et al., 2017). Although aircraft *in situ* instruments can measure $N_d$ directly, these measurements are limited to specific regions and time periods during field campaigns. Collecting a large $N_d$ database from these measurements is a challenging task, making it difficult to statistically study factors that influence the spatial and temporal variations of $N_d$ and ACI processes across different climate zones and atmospheric thermodynamic conditions. Ground-based and space-borne remote sensing techniques provide continuous observations of clouds and aerosols across different regions, and the latter includes global scales. Remote sensing measurements have been widely used to retrieve aerosol and cloud properties including $N_d$. Grosvenor et al. (2018) comprehensively reviewed passive satellite remote sensing retrievals of $N_d$ from the retrieved cloud optical depth, cloud droplet effective radius ($r_e$), and cloud-top temperature. They concluded that satellite $N_d$ retrievals could achieve a relative uncertainty of 78% at the pixel-level for single-layer warm stratiform and optically thick clouds. Ground-based $N_d$ retrievals have higher temporal and spatial resolutions than satellite measurements. By taking advantage of more reliable retrievals of liquid water path (LWP) from passive microwave radiometers, ground-based $N_d$ retrievals usually use cloud optical and LWP instead of $r_e$ in the retrieval algorithms. These remote sensing data provide invaluable information for statistically studying ACI processes and have been used to validate and improve cloud representations in climate models (McComiskey et al., 2009; Rosenfeld et al., 2019; McCoy et al., 2020).



Passive remote sensing $N_d$ retrievals, such as noted above, commonly rely on reflected or transmitted sunlight measured
from spaceborne and ground-based remote sensors, respectively. Therefore, these retrievals are limited to single-layer and
optically thick clouds under conditions when the Sun is high in the sky. These limitations can be alleviated by using active
remote sensing measurements. Active remote sensors transmit electromagnetic waves at a specific visible, infrared, or
microwave wavelength and receive reflected signals from the atmosphere in a narrow field-of-view. Therefore, active remote
sensing measurements can be used for cloud property retrievals anytime (i.e., including nighttime) and under much broader
atmospheric conditions (e.g., beneath cirrus cloud decks).

Ground-based active remote sensing $N_d$ retrievals use either the cloud radar reflectivity factor ($Z$) or lidar extinction
coefficient ($\beta_e$) profiles together with microwave radiometer-retrieved LWP. A monomodal droplet size distribution (DSD)
is usually assumed to connect these measured quantities. Radar-based $N_d$ retrievals use the relationships between $Z$, liquid
water content (LWC), DSD, and $N_d$ (Dong et al., 1998; Mace and Sassen 2000; Wu et al., 2020). Since $Z$ is proportional to
the sixth power of the DSD, radar-based $N_d$ retrievals are very sensitive to the assumed DSD, and it is challenging to retrieve
$N_d$ under drizzling conditions. Recently, lidar-based $N_d$ retrievals have been developed by synergizing multiple instruments
similar to the radar-based retrievals (Boers et al., 2006, Martucci and O'Dowd, 2011; Snider et al., 2018; Zhang et al., 2019)
by using dual-field-of-view lidar extinction profiles (Schmidt et al., 2013) or by using lidar multiple scattering measurements
(Donovan et al., 2015). Since lidar measurements are proportional to the second moment of cloud DSD, lidar-based $N_d$
retrievals are more sensitive to $N_d$ than radar-based methods and have the potential to provide more accurate retrievals.

In the past decade, there has been significant progress in developing $N_d$ retrieval algorithms; however, the validation of these
algorithms against *in situ* measurements is still inadequate. Most $N_d$ retrieval methods were developed and tested under
specific conditions, making it crucial to evaluate their performance against *in situ* measurements from different locations and
cloud conditions to understand better their uncertainties and to confidently extend these algorithms. The Department of
Energy (DOE) Atmospheric Radiation Measurements (ARM) Aerosol and Cloud Experiments in the Eastern North Atlantic
(ACE-ENA) field campaign offers an excellent opportunity to validate different $N_d$ retrieval algorithms under the same range
of cloud conditions. The ACE-ENA campaign (Wang et al., 2022) collected comprehensive data sets from the ARM Eastern
North Atlantic (ENA) site, where the ARM Aerial Facility (AAF) research aircraft made *in situ* measurements over the
Azores where the ENA atmospheric observatory routinely makes measurements from state-of-the-art remote sensing
instruments. The flights during ACE-ENA were designed to take full advantage of the synergy between aircraft *in situ*
measurements and ARM ground-based remote sensing observations. In this study, four $N_d$ retrieval algorithms are evaluated,
considering their potential for operational applications and ease of use across different locations. These methods include two
lidar-based retrievals similar to Snider et al. (2018), a radar-based retrieval similar to Wu et al. (2020a), and the $N_d$ retrieval
from    the    ARM    Droplet    Number    Concentration    (NDROP)    Value-Added    Product    (VAP)    available    at



https://www.arm.gov/capabilities/vaps/ndrop. This study evaluates the $N_d$ retrieval algorithms against *in situ* data to enhance our understanding of their uncertainties and extend their application to other locations.

The paper is organized as following: Section 2 presents a brief introduction of the ARM ENA site, the lidar- and radar-based retrieval algorithms, the ARM NDROP VAP, and the ACE-ENA field campaign measurements; Section 3 shows evaluations of $N_d$ retrievals with *in situ* probe measurements during the ACE-ENA field campaign, and a four-year climatology of overcast marine boundary layer (MBL) cloud $N_d$ climatology based on retrievals at the ENA observatory; and Section 4 presents the summary and conclusions.

## 2 Ground-based $N_d$ Retrievals and ACE-ENA Measurements

The lidar-based retrievals, radar-based retrieval, and the ARM NDROP VAP use different remote sensing measurements and algorithms to retrieve $N_d$. Brief descriptions of these methods are presented in sections 2.2-2.4. These retrieval methods use both passive and active remote sensing measurements. We expect the ensemble of these peer-reviewed retrievals for the same cloud to indicate a reasonable range of the retrieved $N_d$. We refined the lidar-based retrieval method in section 2.2.

Then we evaluated assumptions in each retrieval method, and for the first time, compared four different $N_d$ retrievals with *in situ* measurements to evaluate the robustness of their performances.

### 2.1 The ARM ENA Atmospheric Observatory

Established in October 2013, the ARM ENA atmospheric observatory is located on Graciosa Island in the Azores, Portugal, at 39° 5′ 29.76″ N, 28° 1′ 32.52″ W. This region of the northeastern Atlantic Ocean is characterized by the presence of

marine stratocumulus clouds and is subject to diverse meteorological and aerosol conditions (Wood et al. 2015). Consequently, the ARM ENA site presents an ideal opportunity to study the properties of clouds and precipitation in a remote marine environment as well as the response of low clouds to natural and anthropogenic aerosols and meteorological conditions. Facilitating these studies, the ARM ENA atmospheric observatory has been equipped with a large array of advanced instruments, capable of providing high spatial and temporal resolution measurements of the atmospheric state,

aerosols, clouds, precipitation, and radiation budget. These instruments include a variety of aerosol instrumentation, lidars, radars, radiometers, as well as the balloon-borne sounding (SONDE) system. Table 1 lists the key ground-based instruments and their measurements which were used for $N_d$ retrievals in this study.





## 2.2 Lidar-based $N_d$ Retrieval

In this study, Raman Lidar (RL) and Micropulse lidar (MPL) data are used in separate lidar-based retrievals. The method for
retrieving $N_d$ employs the interrelationships among $N_d$, $\beta_e$, $LWC$, and cloud DSD, where $\beta_e$ is the extinction coefficient
(Snider et al., 2017). At an altitude $z$ above the cloud base, $N_d$, $\beta_e$, and $LWC$ can be expressed as functions of the cloud DSD:

$$N_{d,z} = \int_0^\infty n_{d,z} dr \tag{1}$$

$$\beta_{e,z} = Q_{ext}\pi \int_0^\infty n_{d,z} r^2 dr \tag{2}$$

$$LWC_z = \frac{4}{3}\pi\rho_w \int_0^\infty n_{d,z} r^3 dr \tag{3}$$

where $Q_{ext}$ is the extinction efficiency, $r$ is the cloud droplet radius, $n_{d,z}$ is the droplet number concentration within the size
range between $r$ and $r+dr$, and $\rho_w$ is the density of liquid water. Since water droplet sizes are much larger than the lidar laser
wavelength, $Q_{ext} \approx 2$. The cloud droplet effective radius $r_{e,z}$ is defined as:

$$r_{e,z} = \frac{\int_0^\infty n_{d,z} r^3 dr}{\int_0^\infty n_{d,z} r^2 dr} = \frac{3Q_{ext}}{4\rho_w} \frac{LWC_z}{\beta_{e,z}} \tag{4}$$

To establish a connection between the properties that are a function of the second and third moment of the cloud DSD,
respectively $\beta_{e,z}$ and $LWC_z$, previous research has made the assumption that the cloud DSD follows either a Gamma
distribution or lognormal distribution and has a constant spectrum width (Martucci and O'Dowd, 2011; Snider et al., 2017).
Drawing inspiration from the passive remote sensing retrieval algorithms outlined by McComiskey et al. (2009), a parameter
$k$ is introduced to link $\beta_{e,z}$ and $LWC_z$ that is a measure of the width of the cloud DSD. This parameter represents the cube of
the ratio between the volume radius and the effective radius:

$$k = \frac{1}{N_{d,z}} \int_0^\infty n_{d,z} r^3 dr / r_{e,z}^3 \tag{5}$$

To determine $N_d$, the $k$ parameter is assumed to remain constant vertically within the cloud (Brenguier et al., 2011). Through
the analysis of aircraft *in situ* probe measurements from five distinct field experiments, Brenguier et al. (2011) demonstrated
that the $k$ parameter values range from 0.7-0.9, with uncertainties between 10% and 14% across different cloud systems and
various atmospheric conditions. By integrating equations (2), (3), and (5), $N_d,z$ can be derived as a function of $\beta_{e,z}$ and
$LWC_z$:

$$N_{d,z} = \frac{2\rho_w^2}{9\pi k} \frac{\beta_{e,z}^3}{LWC_z^2} \tag{6}$$

The newly derived Equation (6) eliminates the need for assuming a specific DSD shape (e.g., Gamma or lognormal
distribution) that was necessary in previous studies.

To derive $N_d$, the $LWC_z$ in stratiform cloud is typically assumed to be a constant fraction ($f_{ad}$) of its adiabatic value
($LWC_{z,ad}$): $LWC_z = f_{ad} LWC_{z,ad}$. The $LWC_{z,ad}$ profile can be determined from cloud-base temperature and pressure
measurements. By analyzing two years of ground-based remote sensing data set at Leipzig, Germany, Merk et al. (2016)



shows that $f_{ad}$ values are 0.63±0.22. In this study, $f_{ad}$ is calculated as the ratio of the retrieved LWP from the MWRRETv2 VAP (https://www.arm.gov/capabilities/science-data-products/vaps/mwrretv2) to the LWP calculated from the adiabatic

LWC profile. The MWRRETv2 VAP retrieves LWP from microwave radiometer brightness temperature measurements at 23.8, 31.4, and 90 GHz using the retrieval algorithm developed by Turner et al. (2007). The third channel at 90 GHz provides additional sensitivity to liquid water enabling an LWP uncertainty of ± 10-15 g/m$^2$ (Cadeddu et al., 2013).

Advanced lidar systems, such as the RL and High Spectral Resolution Lidar, are absolutely calibrated by referencing to

molecular scattering. These systems offer reliable estimates of particulate backscatter and extinction coefficients by solving the lidar equation (Thorsen and Fu, 2015; Marais et al., 2016). Our RL retrieval use the RL-estimated $\beta_{e,z}$ from the ARM Raman Lidar Profiles – Feature detection and Extinction (RLPROF-FEX) VAP (https://www.arm.gov/capabilities/science-data-products/vaps/rlprof-fex), which computes $\beta_{e,z}$ using the algorithms developed by Thorsen et al. (2015) and Thorsen and Fu (2015). However, due to the weak strength of the Raman scattering compared to the elastic scattering, noise poses a

considerable challenge for the extinction coefficient retrieval. To enhance the signal-to-noise (SNR) ratio, the fine-resolution RL data at 10 s temporal and 7.5 m vertical resolution are aggregated coarser resolutions of 2 min and 30 m, respectively. While enhancing the SNR, this coarser resolution RLPROF-FEX $\beta_{e,z}$ may introduce additional uncertainty in $N_d$ retrievals for broken clouds. It is important to note that advanced lidar systems are more costly and, as a result, are not widely available.


In contrast, elastic-scattering lidars, such as the MPL and ceilometer, are available at all ARM observatories and numerous locations worldwide including the MPLNET and Cloudnet (Welton et al., 2001; Illingworth et al., 2007). These instruments provide high temporal and vertical measurements of the strong elastic scattering from atmospheric particles. However, elastic-scattering lidar measurements cannot be directly used to derive particulate backscatter and extinction coefficients

since there is only one lidar equation (measurement) for these two variables, i.e., one equation two unknowns. This issue is often addressed using the lidar extinction-to-backscatter ratio ($S$), which represents the relationship between particulate backscatter and extinction coefficients. Once $S$ is determined, the lidar $\beta_{e,z}$ can be inverted from the MPL backscatter intensity measurements by analytically solving the lidar equation using the inversion method developed by Klett (1981) and Fernald (1984). For liquid cloud droplets, $S$ is approximately 18.8 (O'Connor et al., 2004; Thorsen and Fu 2015). To account

for multiple scattering from liquid droplets, a multiple-scattering correction scheme developed by Hogan (2008) is applied. Sarna et al. (2021) demonstrated that, after all corrections to elastic-scattering lidar signals, the inversion method could obtain $\beta_{e,z}$ with an error less than 5% within 90 m above cloud base at the lidar wavelength of 355 nm. We assess the sensitivity and reliability of lidar-based $N_d$ retrievals using both RL- and MPL-estimated $\beta_{e,z}$.



It should be noted that $N_d$ retrievals at cloud base ($Z_{cb}$) are adversely impacted by noise introduced by turbulent mixing. Entrainment mixing may cause $LWC_{cb}$ to deviate significantly from the adiabatic value, resulting in considerable differences between the retrieved $N_{d,cb}$ and $N_{d,z}$ above the $Z_{cb}$. Furthermore, lidar can only penetrate the low portion of the liquid cloud due to the strong attenuation by liquid droplets. The signal becomes fully attenuated when the optical depth reaches ~3 which corresponds to 100 to 300 m above the $Z_{cb}$. Consequently, our retrievals use $\beta_{e,z}$ and $LWC_z$ only within the range

between $Z_{cb}$ + 30 m and $Z_{cb}$ + 90 m. For lightly drizzling maritime stratocumulus clouds, such as those with the column maximum radar reflectivity ($Z_e$) < 0 dBZ, the contribution of drizzle particles to lidar extinction is negligible compared to that from liquid droplets; thus, the lidar-based retrievals can still be employed. Based on equation (4) and the assumptions that the cloud maintains a constant fraction of its adiabatic value and $N_d$ remains vertically constant, $r_e$ for the rest of the cloud layer can be estimated.


Using a similar lidar-based $N_d$ retrieval approach, Snider et al. (2017) discovered that, in general, the lidar-based $N_d$ retrievals were smaller than *in situ* probe measurements during the VAMOS Ocean–Cloud–Aerosol–Land Study (VOCALS) Regional Experiment over the southeastern Pacific (Wood et al. 2011). It is worth noting that Snider et al. (2017) used the adiabatic LWC lapse rate without considering the subadiabaticity, which results in an overestimation of $LWC_z$ and

consequently an underestimation of $N_d$ based on equation (6). Therefore, in the present study, the bias of the $N_d$ retrieval should not be as large since we consider cloud subadiabaticity.

## 2.3 Radar-based $N_d$ Retrieval

Obtaining $N_d$ values from radar reflectivity poses challenges due to the frequent presence of drizzle within MBL clouds, which subsequently contributes significantly to the measured radar reflectivity (Zhu et al., 2022). Wu et al. (2020a) recently

developed a method to separate drizzle and cloud droplet contributions to the measured radar reflectivity while simultaneously retrieving cloud and drizzle microphysical properties, including $N_d$, in precipitating MBL clouds.

To distinguish between drizzle and cloud droplet contributions, moving downward from the cloud top they identify the height at which $Z_e$ becomes larger than -15 dBZ, demarcating the level of drizzle initiation and the point above which the

measured $Z_e$ is exclusively attributed to cloud droplets. While it is convenient to use this threshold, it should be noted that a number of recent studies demonstrate that drizzle having significantly lower reflectivity that -15 dBZ can be observed within stratocumulus clouds (Kollias et al., 2011; Luke at al., 2013; Zhu et al., 2022). The cloud contribution to $Z_e$ at the cloud base is calculated as the difference of $Z_e$ from the radar range gates above and below cloud base. Subsequently, they construct the cloud radar reflectivity ($Z_c$) by assuming a linear increase in cloud liquid water content ($LWC_c$) with height above cloud base

(and thus a linear increase in $\sqrt{Z}$ if $N_d$ is invariant with height). By assuming that the cloud droplet particle size distribution



follows a lognormal distribution with a logarithmic width of $\sigma_x$, the relationship between $N_d$, $LWC_c$, and $Z_c$ can be expressed as:

$$LWC_c = \frac{\pi}{6}\exp\left(-4.5\sigma_x^2\right)\sqrt{N_d Z_c} \tag{7}$$

The logarithmic width $\sigma_x$ is set to 0.38, as suggested by Miles et al. (2000). To determine $N_d$, equation (7) is further

constrained by the cloud LWP, derived from the difference between the MWRRETv2 (total) LWP and the calculated drizzle water path, which is obtained from the retrieved drizzle water content profile. Subsequently, the $r_e$ profile is derived from $N_d$ and the $LWC_c$ profile.

To mitigate the impact of MWRRETv2 LWP uncertainties, cloud microphysical property retrievals were smoothed to a

temporal resolution of 1 min. A sensitivity analysis conducted by Wu et al. (2020a) revealed that the retrieved $N_d$ values are not sensitive to the selection of the radar reflectivity threshold of -15 dBZ. Using aircraft measurements from the ACE-ENA field campaign as a benchmark, the median $N_d$ retrieval error is approximately ~ 35%.

**2.4 The ARM NDROP VAP**

The $N_d$ retrieval method employed by the ARM NDROP VAP uses the relationship between LWP, cloud optical depth ($\tau$),

cloud DSD, and $N_d$. Following Lim et al. (2016), the layer-mean $N_d$ can be expressed as:

$$N_d = \left[\frac{2^{-5/2}}{k}\right]\left[\frac{3\pi}{5}Q_{ext}\right]^{-3}\left[\frac{3}{4\pi\rho_w}\right]^{-2}\tau^3 LWP^{-5/2}(f_{ad}c_w)^{1/2} \tag{8}$$

As both $\tau$ and LWP represent vertical integrals through the entire cloud layer, Brenguier et al. (2011) propose using the cloud system $k^*$ parameter in place of $k$ in equation (8). In the case of a linearly stratified cloud with constant $k$ within the cloud, $k^*$ can be derived as $k^* = 0.864\,k$. The NDROP VAP adopts a $k^*$ value of 0.74, as recommended by Brenguier et al.

(2011) (Riihimaki et al., 2021).

The adiabatic LWC lapse rate, $c_w$, can be calculated using cloud-base temperature and pressure from the ARM INTERPSONDE VAP (https://www.arm.gov/capabilities/vaps/interpsonde) (Jensen and Toto 2016). $\tau$ is available from the ARM MFRSRCLDOD VAP (https://www.arm.gov/capabilities/vaps/mfrsrcldod), which retrieves $\tau$ for overcast liquid

clouds from multifilter rotating shadowband radiometer (MFRSR) measurements using the retrieval algorithm developed by Min and Harrison (1996) (Turner et al., 2021). This algorithm employs the transmitted irradiance at 415 nm from the MFRSR, so the retrieved $\tau$ is available only during daytime. Analyses show that $\tau$ from the MFRSRCLDOD VAP has uncertainties ranging from 0.5 to 2.5 (Turner et al., 2021). LWP is available from the ARM MWRRETv2 VAP. Since both the $\tau$ and LWP retrievals have significant relative uncertainties for optically thin clouds, this retrieval approach should be

applied for overcast, optically thick liquid clouds.



Lim et al. (2016) evaluated $N_d$ values retrieved using this approach by comparing them to aircraft *in situ* probe measurements obtained during the Routine ARM Aerial Facility (AAF) Clouds with Low Optical Water Depths (CLOWD) Optical Radiative Observations (ROCORO) field campaign at the ARM Southern Great Plains (SGP) site (Vogelmann et al.,

2012). Their findings indicate that the retrieved $N_d$ values are substantially larger than the *in situ* measurements. This discrepancy may be attributed to the fact that clouds sampled during the ROCORO campaign often exhibited small LWPs. Consequently, NDROP retrievals still require evaluation under optically thick cloud conditions.

For passive remote sensing retrievals, the layer-mean $r_e$ ($r_{em}$) between the cloud layer top and base can be determined using the relationship among $r_{em}$, $\tau$, and *LWP*:

$$r_{em} = \frac{3LWP}{2\rho\tau} \qquad (9)$$

$r_{em}$ is available from the ARM Cloud Optical Properties from the Multifilter Shadowband Radiometer (MFRSRCLDOD) VAP (https://www.arm.gov/capabilities/science-data-products/vaps/mfrsrcldod). We note that Chiu et al. (2012) finds better retrieval results by replacing the '3/2' factor in equation (9) with '9/5', where the latter factor assumes that $N_d$ is approximately constant and the LWC increases linearly with height (Wood and Hartmann, 2006). Were '9/5' used in the

NDROP VAP, the $r_{em}$ would be 20% larger than those values reported here. To compare $r_e$ retrievals from different approaches and with *in situ* measurements, we use $r_{em}$ for comparisons for the rest of the discussion following previous studies (Chiu et al., 2012; Grosvenor et al.,2018). $r_{em}$ is derived by averaging $r_e$ at each layer between the cloud top and base from lidar- and radar-based retrievals, and *in situ* measurements.

### 2.5 ACE-ENA *In Situ* Measurements

The ACE-ENA field campaign deployed the ARM AAF Gulfstream-159 (G-1) research aircraft over the Azores during the two intensive operational periods (IOPs) in early summer 2017 (June to July) and winter 2018 (January to February). The G-1 was equipped with a range of *in situ* sensors, enabling comprehensive measurements of aerosol particles, cloud droplets, precipitation, and atmospheric conditions. Cloud probes particularly relevant to $N_d$ measurements include the Fast-Cloud Droplet Probe (FCDP) and the Cloud and Aerosol Spectrometer (CAS). The FCDP measures cloud droplets in the diameter

size range of 1.5-50 μm with temporal resolution of 1 or 0.1 s. The CAS provides measurements of aerosol or cloud droplets in the 0.5-50 μm diameter size range with a temporal resolution of 1 s. Given the different particle size ranges measured by the various probes, we used *in situ* $N_d$ data for the particle size between 3-50 μm. It is noted that although *in situ* probes provide reliable $N_d$ measurements, they also have uncertainties ranging from 10-30% as presented by Baumgardner et al. (2017). Therefore, we include both FCDP and CAS measurements for evaluating $N_d$ retrievals.


During the ACE-ENA campaign, the G-1 aircraft conducted both vertical profiling flights and horizontal flights at physically important levels, such as near the ocean surface, just below clouds, within clouds, as well as at and above the cloud top (Wang et al., 2022). These flights were specifically designed to maximize synergy between G-1 aircraft measurements and





ENA ground-based remote sensing observations, offering an ideal dataset for evaluating $N_d$ retrievals. Most G-1 flights

employed an L-shaped pattern, including both upwind and crosswind legs at different altitudes, with the L "corner" over the

ENA site. Additionally, four G-1 flights used a "Lagrangian drift" pattern, starting upwind of the ENA site and performing

crosswind measurements while drifting with the prevailing boundary layer winds (Wang et al., 2022). In total, 39 flights

were conducted during the two IOPs.

Among those flights, 12 flight days featuring multiple in-cloud flight legs under single-layer stratiform cloud conditions

were selected for this study. Heavily drizzling stratocumulus flight days were excluded. Table 2 provides the date, in-cloud

flight time, and cloud conditions for the 12 flight days. In-cloud measurements are defined as those when the FCDP-

measured $N_d$ values are larger than 10 cm$^{-3}$. Approximately 11 total hours of in-cloud flight measurements were used to

evaluate the $N_d$ retrievals. Figure 1 illustrates cloud properties of the 12 selected flight days, including fractional sky cover,

cloud-base height, LWP, and column-maximum $Z_e$ ($Z_{e\_max}$) derived from the ENA ground-based remote sensing

observations. The box-and-whisker plots display the 5th, 25th, 50th, 75th, and 95th percentiles. Of the 12 selected flight

days, nine have overcast cloud conditions and three have broken cloud conditions (6/28/2017, 7/8/2017, 2/12/2018). These

three broken cloud days had among the smallest LWPs, as shown in Figure 1c. Cloud-base heights ranged from 0.5 to 1.5

km with variations often smaller than 0.3 km on a given flight day. Overall, these clouds had LWPs less than 200 g/m$^2$ and

$Z_{e\_max}$ values smaller than 0 dBZ, which are typical for marine low-level clouds.

## 3 Results and Discussions

### 3.1 Evaluation of Retrieval Assumptions

In the lidar-based, radar-based, and NDROP VAP $N_d$ retrievals, several assumptions are made regarding the vertical $N_d$

variation, the $k$ and $k^*$ parameter, and the LWC profile, as described in sections 2.2 and 2.3. We test these assumptions in

this section. Figure 2 presents statistics of these cloud properties from *in situ* and ground-based measurements during the 12

selected flight days. For example, Figure 2a shows that the mean $N_d$ normalized by the flight average is close to 1, with

standard deviations of approximately 0.4 though the cloud layer, which supports the assumption that $N_d$ can be treated as

constant within the cloud layer.

The $k$ parameter is assumed to be vertically constant. Some previous studies find that the $k$ parameter increases with height

(Brenguier et al., 2011), while others suggest that the $k$ parameter can either increase or decrease with height (Pawlowska et

al., 2006; Painemal and Zuidema 2010). Our analysis shows that the mean $k$ parameter remains essentially constant with

height (Figure 2b). The probability distribution functions (PDF) of the $k$ and $k^*$ parameters (Figure 2c) reveal that $k$ ($k^*$)

ranges between 0.6 (0.5) and 1.0 (0.86), with a mean value of 0.86 (0.74) and a standard deviation of 0.10 (0.09). Since $N_d$ is

inversely proportional to $k$ ($k^*$) as shown in equation (6) and (8), an uncertainty of 0.10 in $k$ value alone could cause an



uncertainty of ~12% in the retrieved $N_d$ value from the uncertainty propagation analysis. The lidar-based $N_d$ retrievals in this study use a $k$ value of 0.86. We note that the $k^*$ value of 0.74 used in the NDROP VAP is well justified. As the $k$ value of 0.86 corresponds exactly to the recommended $k^*$ value of 0.74 by Brenguier et al. (2011), where their value is based on data from five field program locations, it suggests that a $k$ value of 0.86 might be more broadly applicable for lidar-based $N_d$

retrievals of boundary layer clouds at other locations.

As aircraft *in situ* probes are unable to provide continuous cloud-base height measurements, and the $LWC_{ad}$ or $c_w$ profile is sensitive to cloud-base height, it is challenging to determine $f_{ad}$ and its vertical variations within a cloud. Instead, we use the ratio of the MWRRETv2 LWP to the computed adiabatic LWP to calculate $f_{ad}$. As seen in Figure 2d, the adiabatic LWPs are

generally larger than MWRRETv2 LWPs, especially when the LWP is above 150 g/m$^2$, but they correlate well. The PDF of $f_{ad}$ shows that $f_{ad}$ has a mean value of 0.76, with a standard deviation of 0.42 (Figure 2e). Since cloud LWP should not exceed the adiabatic LWP, $f_{ad}$ is set to 1 when it is larger than 1. Those values, and possibly those at the extreme lower end of $f_{ad}$, appear to be affected primarily by uncertainties in cloud thickness for thin clouds (< 200 m) and by uncertainties in low MWRRETv2 LWPs (< 75 g/m$^2$), based on scatter plots of $f_{ad}$ vs. these respective properties (not shown). On the other

end, when the LWP is above ~150 *g/m$^2$*, the cloud could contain drizzle. The current MWRRETv2 retrieval does not account for drizzle scattering effects at frequencies above 90 GHz, which could cause overestimation of the LWP by 10-15%, as outlined in the study by Cadeddu et al. (2020). We did not implement corrections to this bias due to two reasons: firstly, there are currently no reliable methods to correct such bias; secondly, we removed strong drizzling stratocumulus cases by excluding clouds with $Z_{e\_max}$ larger than 0 dBZ as discussed in section 2.

**3.2 Evaluation of $N_d$ Retrievals**

For convenience, we label $N_d$ $(r_{em})$ retrievals from the MPL, RL, KAZR radar measurements, and from the NDROP (MFRSRCLDOD) VAP as $N_{d\_mpl}(r_{em\_mpl})$, $N_{d\_rl}(r_{em\_rl})$, $N_{d\_radar}(r_{em\_radar})$, $N_{d\_vap}(r_{em\_vap})$, respectively, and *in situ* measured $N_d$ $(r_{em})$ from FCDP and CAS as $N_{d\_FCDP}(r_{em\_FCDP})$, $N_{d\_cas}(r_{em\_CAS})$. Figure 3 shows an example of ground-based remote sensing measurements and $N_d$ and $r_e$ retrievals on January 26, 2018. The cloud is a typical stratiform MBL cloud with a cloud-base

height of ~1.1 km, and a cloud-top height of ~1.5 km. The cloud system persisted for more than 55 hours from ~5:00 UTC January 25 to ~12 UTC January 27 (full period not shown in Figure 3). From the mean sea level pressure distribution (Fig. S1), the Azores high was located to the northeast of the Azores. Near-surface winds were south to southeast across the ENA observatory. The synoptic environment created a strong stable boundary layer condition that was favorable for the maintenance of marine boundary layer stratocumulus. From Figure 3a and b, large $\beta_e$ and its rapid attenuation indicate the

presence of the liquid layer. Figure 3c shows radar reflectivity up to -20 dBZ below the liquid layer, indicating that drizzle frequently forms and falls out of the liquid layer. The mean MWRRETv2 LWP ($LWP_{mwr}$) and calculated adiabatic cloud LWP ($LW_{ad}$) are 107 g/m$^2$ and 119 g/m$^2$, respectively. Figure 3d shows that $LWP_{ad}$ and $LWP_{mwr}$ correlate very well and are



close in magnitude, indicating the cloud is nearly adiabatic. Retrieved $N_{d\_mpl}$, $N_{d\_rl}$, $N_{d\_radar}$, $N_{d\_vap}$ are shown in Figure 3e. For

this case, $N_{d\_mpl}$ and $N_{d\_radar}$ have a similar magnitude at ~ 50 cm$^{-3}$ but are smaller than $N_{d\_rl}$ and $N_{d\_vap}$. Derived $r_{em\_mpl}$, $r_{em\_rl}$,

$r_{em\_radar}$, and $r_{em\_vap}$ are very close at ~ 10.5 μm as shown in Figure 3f.

This case is one of the four "Lagrangian drift" flights during the entire ACE-ENA field campaign. The prevailing boundary

layer winds were south- to southeastward. Boundary layer wind speeds were generally less than 10 m/s, based on radiosonde

measurements at 11:30 UTC at the ENA observatory (Figure 4a). The G-1 aircraft took off at approximately 11:05 UTC

upwind of the ENA observatory and landed at around 15:00 UTC (Figure 4). During the four-hour flight, the G-1 aircraft

made about 1 hour and 37 minutes of in-cloud measurements, including several horizontal legs just below cloud top, within

the cloud layer, and just above cloud base, as well as several spirals. Satellite imagery from the Moderate Resolution

Imaging Spectroradiometer (MODIS) shows that closed-cellular stratocumulus clouds dominated the region (Figure 4b).

Due to the continuous movement of the G-1 aircraft near the ENA observatory, establishing direct one-to-one comparisons

between ground-based retrievals and aircraft *in situ* measurements is challenging. Instead, we evaluate the PDFs of $N_d$

retrievals against those from the aircraft *in situ* measurements. Figure 5a shows the comparison of ground-based $N_d$ retrievals

$N_{d\_mpl}$, $N_{d\_rl}$, $N_{d\_radar}$, and $N_{d\_vap}$ during the time of the concurrent aircraft flight against *in situ* FCDP ($N_{d\_FCDP}$) and CAS

($N_{d\_CAS}$) measurements for the case on January 26, 2018. It should be noted that measurements from the two *in situ* probes

show slightly different $N_d$ distributions. $N_{d\_CAS}$ is generally less than $N_{d\_FCDP}$ with a median of 74 cm$^{-3}$ and a narrower

distribution with a standard deviation of 24 cm$^{-3}$, while $N_{d\_FCDP}$ has a median of 97 cm$^{-3}$ and a standard deviation of 34 cm$^{-3}$.

Among the four $N_d$ retrievals, $N_{d\_mpl}$ shows a very similar distribution to $N_{d\_cas}$, with a median of 73 cm$^{-3}$ and a standard

deviation of 31 cm$^{-3}$. $N_{d\_rl}$ shows a broader distribution with a median of 114 cm$^{-3}$ and a standard deviation of 71 cm$^{-3}$,

probably because the retrieved RL $\beta_e$ has a larger random noise than that of MPL $\beta_e$. $N_{d\_radar}$ has the narrowest distribution,

with a median of 62 cm$^{-3}$ and a standard deviation of 13 cm$^{-3}$. $N_{d\_vap}$ has the greatest retrievals, with a median of 127 cm$^{-3}$

and a standard deviation of 46 cm$^{-3}$.

As $r_e$ changes with distance above cloud base and it is challenging to know instantaneous cloud-base height from aircraft

measurements, it is more difficult to conduct one-to-one comparisons between ground-based $r_e$ retrievals and aircraft *in situ*

measurements. Therefore, we compare PDFs of the $r_{em}$ retrievals against aircraft *in situ* measurements during all in-cloud

penetrations. Figure 5b shows that the median $r_{e\_FCDP}$ and $r_{e\_CAS}$ are almost the same, around 10.4 μm. The median (standard

deviation) of $r_{em\_mpl}$, $r_{em\_rl}$, $r_{em\_radar}$, and $r_{e\_vap}$ are 11.5 μm (1.9 μm), 10.2 μm (2.6 μm), 9.3 μm (0.7 μm), and 10.5 μm (1.1

μm), respectively. Although median $r_{em}$ values from different retrieval methods are very close, $r_{em\_mpl}$ *and* $r_{em\_rl}$ have broader

distributions than $r_{em\_radar}$ and $r_{e\_vap}$.




Figure 6 presents the comparison of ground-based $N_d$ and $r_{em}$ retrievals against *in situ* FCDP and CAS measurements for the 12 selected flight days. Table 3 displays the median $N_d$ of the 12 selected flight days and their relative differences with respect to $N_{d\_FCDP}$. In accordance with prior studies of cloud microphysical properties (Yeom et al., 2021; Zhang et al., 2021), we consider FCDP measurements as the benchmark. The median $N_{d\_FCDP}$ for the 12 days ranges from 33 to 125 cm$^{-3}$. There

are substantial variations in $N_d$ from *in situ* measurements among the 12 days, with higher $N_d$ observed on 06/30/2017, 07/06/2017, and 07/08/2017, and lower $N_d$ on 07/18/2017, 01/19/2018, and 01/25/2018. This agrees with the analysis in Wang (2022) of all *in situ* $N_d$ measurements during the ACE-ENA field campaign, which reveals that the flight-mean $N_d$ ranges from 20 – 50 cm$^{-3}$, and that summer IOP $N_d$ is generally larger than that of the winter IOP. Encouragingly, ground-based $N_d$ retrievals generally follow the same seasonal variation trend as, shown in Figure 6a.


Between the two *in situ* probe measurements, $N_{d\_FCDP}$ and $N_{d\_CAS}$ show good agreement. The median $N_d$ relative differences of $N_{d\_CAS}$ with respect to $N_{d\_FCDP}$ are smaller than 10% for most flights (Table 3). However, significant differences are observed for several flights, such as on 01/19/2018 and 02/07/2018, when the median $N_d$ relative differences are larger than 40%. $N_{d\_mpl}$ compares well with *in situ* probe measurements, with the median $N_d$ relative differences in $N_{d\_mpl}$ with respect to

$N_{d\_FCDP}$ ranging from 9% to 89%. Interestingly, Figure 6a reveals that $N_{d\_mpl}$ overestimates $N_d$ during the summer IOP but underestimates $N_d$ during the winter IOP, partially because the $k$ parameter values were smaller (larger) during the summer (winter) IOP than the default value of 0.86 used in the retrieval algorithms (Figure S2). $N_{d\_rl}$ compares well with *in situ* probe measurements for overcast clouds but significantly underestimates $N_d$ for broken clouds (06/28/2017, 07/08/2017, and 02/12/2018), which is likely due to the coarse temporal resolution of RLPROF-FEX extinction data. Similar to the

01/26/2018 case, $N_{d\_radar}$ values for other flight days consistently have a narrower range and are generally smaller than *in situ* probe measurements. $N_{d\_vap}$ considerably overestimates $N_d$ for either broken clouds or when clouds have low LWPs, such as on 06/21/2017, 06/28/2017, and 07/08/2017. For overcast clouds with LWPs greater than ~25 g/m$^2$, $N_{d\_vap}$ compares well with *in situ* probe measurements. Overall, retrieved $N_d$ have a larger spread and poorer comparison with *in situ* probe measurements during the summer IOP than that of the winter IOP, likely because more broken low-level clouds are present

during summer at the ENA observatory (Figure 1a).

Figure 6b reveals significant differences in $r_e$ between the two IOPs, with smaller $r_e$ values during the summer IOP and larger $r_e$ values during the winter IOP. This is in line with the differences in $Z_{e\_max}$ between two IOP as shown in Figure 1d, since $Z_{e\_max}$ is highly sensitive to the presence of large particles. *In situ* probe-derived $r_e$ values are very close to each other,

with differences between $r_{em\_FCDP}$ and $r_{em\_CAS}$ being less than 1 μm for all the 12 selected flight days (Table S1). Retrieved $r_{em\_mpl}$, $r_{em\_rl}$, $r_{em\_radar}$, and $r_{em\_vap}$ all correspond well with the $r_{em\_FCDP}$ variations. $r_{em\_mpl}$ values are slightly larger than $r_{em\_FCDP}$, with absolute differences usually within 2 μm. This is likely because $r_{em\_mpl}$ is calculated assuming a constant subadiabatic LWC profile, leading to a linear increase in $r_e$ from cloud base to cloud top. In reality, cloud $r_e$ increases above cloud base but decreases slightly at cloud top due to entrainment mixing of dry air (Wang et al., 2022). $r_{em\_rl}$ compares well



with $r_{em\_FCDP}$ for most cases but is significantly larger than $r_{em\_FCDP}$ for flight days when the retrieved $N_{d\_rl}$ values are considerably smaller than $N_{d\_FCDP}$ due to broken clouds and the coarse temporal resolution of the RL extinction data. $r_{e\_radar}$ values are also within 2 μm of $r_{e\_FCDP}$ , which can be either larger or smaller. $r_{e\_vap}$ compares well with $r_{e\_FCDP}$ when the retrieved $N_{d\_vap}$ are larger than $N_{d\_FCDP}$, but it is larger than $r_{e\_FCDP}$ when the retrieved $N_{d\_vap}$ values compare well with $N_{d\_FCDP}$.

### 3.3 Implementing N$_d$ Retrievals to multiple years of ENA Data

A significant advantage of ground-based N$_d$ retrievals is their applicability to long-term, continuous, and high temporal resolution remote sensing measurements, facilitating process-level understanding of cloud microphysical properties and their climatology. The N$_d$ retrievals are applied to four years of ground-based remote sensing measurements of overcast MBL clouds at the ENA observatory between 2016 and 2019. MBL clouds are identified as those with base heights lower than 4

km above sea level (ASL). Considering the limitation of RL and NDROP retrievals, we selected single-layer overcast MBL cloud systems that persist longer than 20 minutes with a concurrent TSI fractional sky cover greater than 95% and an LWP greater than 25 g/m². To avoid heavily precipitating cloud systems, we excluded clouds with $Z_{e\_max}$ larger than 0 dBZ. Since the RL data and retrievals have the coarsest temporal resolution of 2 min, other retrievals were subsampled to the same temporal resolution as RL data. In total, approximately 245,000 retrieved N$_d$ and $r_e$ data samples were collected.


Figure 7a displays the monthly occurrence of overcast MBL clouds at the ENA observatory meeting the above-stated criteria. The annual mean occurrence of these clouds is approximately 0.26 with higher monthly mean occurrences in June and July, and a lower occurrence during December. The mean MBL cloud occurrence and its seasonal variations align closely with those of the low-level cloud presented in Wu et al. (2020b), which used a similar dataset to study MBL cloud

and drizzle properties at the ENA observatory but for cloud top height below 3 km. Monthly N$_d$ statistics are shown in Figure 7b. The annual median $N_{d\_mpl}$, $N_{d\_rl}$, $N_{d\_radar}$, and $N_{d\_vap}$ are approximately 79.7, 75.9, 54.4, and 116.9 cm$^{-3}$, respectively. As with the evaluations for the ACE-ENA field campaign, $N_{d\_vap}$ at the ENA observatory are consistently larger than other retrievals. Lim et al. (2016) suggested that unrealistically high $N_{d\_vap}$ over 2000 cm$^{-3}$ generally occur when LWP is low. By limiting retrievals to only MBL systems with LWP greater than 25 g/m², we do not find $N_{d\_vap}$ larger than 500 cm$^{-3}$.

However, the systematically larger $N_{d\_vap}$ compared to other retrievals indicates that cloud optical depth retrievals might also be biased by off-zenith clouds, which are not considered in the cloud optical depth retrievals. $N_{d\_mpl}$ and $N_{d\_rl}$ are generally very close to each other, suggesting that cloud droplet particulate extinction inversion using either the Fernald method or RL data is reasonably reliable. $N_{d\_radar}$ compares well with lidar-based retrievals and has the narrowest distributions each month and the smallest monthly variations. All retrievals show slightly seasonal N$_d$ variations with higher N$_d$ during the summer

season and lower N$_d$ during the winter season, consistent with the N$_d$ differences between the summer IOP and winter IOP during the ACE-ENA field campaign, as discussed in section 3.2. Wang et al. (2022) suggested that N$_d$ is positively





correlated to the boundary layer accumulation mode aerosol concentration, but the ratio of summer to winter $N_d$ is smaller than the seasonal variations of accumulation mode aerosol concentration. Figure 7c shows the monthly distributions of cloud condensation nuclei ($N_{ccn}$) at the supersaturation of 0.1% from the ARM CCN counter (CCN-100) of the surface aerosol observing system (AOS). $N_{ccn}$ has similar seasonal variations as $N_d$ with larger values in June and July, and smaller values in December, but its seasonal variations are much larger than those of $N_d$, consistent with the finding of Wang et al. (2022).

Figure 7d presents the monthly distributions of retrieved $r_{em}$ values. The annual median $r_{em\_mpl}$, $r_{em\_rl}$, $r_{em\_radar}$, and $r_{em\_vap}$ are 14.5, 13.8, 10.4, and 11.7 μm, respectively. Both $r_{em\_mpl}$ and $r_{em\_rl}$ are slightly larger than $r_{em\_radar}$ and $r_{em\_vap}$, due to the assumption of a constant subadiabatic LWC profile when calculating $r_{em\_mpl}$ and $r_{em\_rl}$, as discussed in section 3.2. Also note that the Wu et al. (2020a) method retrieves cloud and drizzle drop size separately and that $r_{em\_radar}$ is the effective radius solely for cloud droplet and does not account for drizzle particle size. Thus, the smaller $r_{em\_radar}$ with respect to other retrievals is expected. While $r_{em\_radar}$ and $r_{em\_vap}$ do not exhibit significant monthly variations, $r_{em\_mpl}$ and $r_{em\_rl}$ are slightly smaller in June and July and slightly larger in November and December, displaying an opposite seasonal variation pattern compared to that of the $N_{d\_mpl}$ and $N_{d\_rl}$ in Figure 7b. Figure 7e illustrates the monthly statistics of $Z_{e\_max}$, which shares a similar seasonal variation pattern with $r_{em\_mpl}$ and $r_{em\_rl}$ in Figure 7d, reinforcing the observed $r_{em\_mpl}$ and $r_{em\_rl}$ seasonal variation pattern.

## 4 Summary

Remote sensing techniques offer extensive cloud properties for studying ACI processes and validating climate model simulations. Validating $N_d$ retrieval algorithms against *in situ* probe measurements is needed to understand their uncertainties. The ARM ACE-ENA field campaign offers a unique opportunity to validate four different $N_d$ ground-based retrieval algorithms, which use ENA atmospheric observatory data, against G-1 research aircraft observations, which flew over the Azores during intensive IOPs in early summer 2017 and winter 2018. Twelve flight days under single-layer stratiform low-level cloud conditions were selected, with six days in the summer IOP and six days in the winter IOP. Approximately 11 total hours of in-cloud flight measurements were used to evaluate $N_d$ retrievals.

Several assumptions used in the retrieval algorithms were assessed or characterized.

- *Cloud DSD Shape*: For the lidar-based $N_d$ retrieval, we demonstrate in equation (6) that using the $k$ parameter can eliminate the need to assume a shape of the cloud DSD (e.g., Gamma or lognormal distribution). The $k$ parameter is the cube of the ratio of the volume radius to the effective radius ($r_e$) of the cloud droplets, representing the width of cloud DSD.
- *Constant $N_d$ with height:* Aircraft *in situ* measurements confirm that $N_d$ can be treated as constant through the cloud layer for stratiform MBL clouds, with the mean $k$ parameter remaining constant with height. The $k$ value ranges



between 0.6-1.0 with a mean of 0.86, which is very close to $k$ values at other geographic locations (Brenguier et al., 2011).

- *Treating Subadiabatic LWC:* The ratio of the retrieved LWP from the MWRRETv2 VAP divided by the LWP calculated from the adiabatic LWC profile is used to estimate the subadiabaticity fraction, $f_{ad}$. The mean value of $f_{ad}$ is 0.76 at the ENA observatory during the ACE-ENA campaign period.

Retrieved $N_d$ ($N_{d\_mpl}$, $N_{d\_rl}$, $N_{d\_radar}$, $N_{d\_vap}$) and cloud-layer-mean $r_e$ ($r_{em\_mpl}$, $r_{em\_rl}$, $r_{em\_radar}$, $r_{em\_vap}$) are evaluated against aircraft *in situ* probe measurements of $N_d$ ($N_{d\_FCDP}$, $N_{d\_CAS}$) and $r_{em}$ ($r_{em\_FCDP}$, $r_{em\_CAS}$). To manage the challenge of direct one-to-one comparisons between ground-based retrievals and aircraft *in situ* measurements, we compare the PDFs of the retrievals with aircraft measurements. Analysis of the *in situ* measurements and retrievals for the 12 flight days reveal the following.

1) There is good agreement in the $N_d$ *in situ* probe measurements, $N_{d\_FCDP}$ and $N_{d\_CAS}$, with the relative differences in the median $N_d$ often being smaller than 10% for most flights (albeit with larger differences in some cases).
2) Ground-based $N_d$ retrievals generally follow the same day-to-day variation of the *in situ* measurements.
3) The assessment of the $N_d$ retrievals with the *in situ* measurements reveals:
   a. $N_{d\_mpl}$ compares well overall with the aircraft measurements, but it overestimates $N_d$ during the summer IOP and underestimates it during the winter IOP;
   b. $N_{d\_rl}$ compares well for overcast clouds but underestimates $N_d$ for broken clouds;
   c. $N_{d\_radar}$ values are consistently smaller and have a narrower range than *in situ* measurements;
   d. $N_{d\_vap}$ overestimates $N_d$ for broken clouds or clouds with low LWPs.
4) There is good agreement in the $r_{em}$ *in situ* probe measurements, $r_{em\_FCDP}$ and $r_{em\_CAS}$. The evaluations of $r_{em}$ show that the retrievals following the variations in $r_{em\_FCDP}$. There is a tendency for $r_{em\_mpl}$ to be slightly larger than $r_{em\_FCDP}$.

These retrieval algorithms are further applied to four years of continuous ground-based remote sensing measurements of overcast MBL clouds at the ENA observatory. Monthly statistics of $N_d$ ($r_{em}$) show slightly seasonal variations with a tendency towards higher (lower) values during the summer season and lower (higher) values during the winter season. $N_{d\_mpl}$ and $N_{d\_rl}$ are generally very close to each other. The $N_{d\_vap}$ is found to be systematically larger than other retrievals, which might arise from the dissimilar fields of view (FOVs) for the cloud optical depth and LWP retrievals, where the former is a hemispheric FOV while the latter is a zenith radiance. $N_{d\_radar}$ compares well with lidar-based retrievals and has the narrowest distributions each month with the smallest monthly variations. $r_{em\_mpl}$ and $r_{em\_rl}$ are found to be slightly larger than $r_{em\_radar}$ and $r_{em\_vap}$.



$N_d$ retrievals evaluated in this study use various remote sensing measurements and employed different retrieval algorithms. Consequently, the ensemble of these retrievals for the same cloud can help us to quantify $N_d$ retrieval uncertainties and identify reliable retrievals, such as when the ensemble of all retrievals has a narrow range (Zhao et al., 2012). Out of the four

retrieval methods, we recommend using the MPL lidar-based method given its good agreement with *in situ* measurements, it has less sensitivity to issues arising from precipitation and low cloud LWP/optical depth, and it has broad applicability by functioning for both day and nighttime conditions. Ground-based $N_d$ retrievals can be used to enhance our understanding of local cloud microphysical processes and can provide long-term verification of spaceborne $N_d$ retrievals that can provide a global dataset needed for validating and improving global climate model simulations of clouds (Bennartz and Rausch 2017)



**Data availability**

The ARM ground-based measurements and ACE-ENA field campaign data used in this study can be downloaded from the ARM data archive site: https://www.archive.arm.gov/discovery/. Ground-based $N_d$ and $r_e$ retrievals are available upon request.

**Author contributions**

Conceptualization, D.Z. and A.M.V.; methodology, D.Z., Z.W., P.W., A.M.V.; software, D.Z.; validation, D.Z.; formal analysis, D.Z., Z.W., P.W., A.M.V.; investigation, D.Z.; resources, D.Z.; data curation, D.Z. and P.W.; writing—original draft preparation, D.Z.; writing—review and editing, all co-authors; visualization, D.Z.; supervision, D.Z.; project administration, D.Z.; funding acquisition, W.I.G. and A.M.V. All authors have read and agreed to the published version of the manuscript.

**Competing interests**

The authors declare that they have no conflict of interest.

**Acknowledgements**

We thank the ACE-ENA field campaign team for their data collection during challenging operations. Data were obtained from the ARM user facility, a U.S. DOE Office of Science user facility managed by the Biological and Environmental Research (BER) program. This research was supported by the DOE ARM and the LASSO program.



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



**Table 1: Ground-based instruments and measurements at the ENA site used in this study.**

| Instrument | Temporal/vertical resolutions | Measured or derived quantities |
|---|---|---|
| **Micropulse Lidar (MPL)** | 10 s/15 m | Lidar backscatter intensity, linear depolarization ratio |
| **Raman Lidar (RL)** | 10 s/7.5 m | Particulate lidar backscatter and extinction coefficient, linear depolarization ratio |
| **Ka-band ARM Zenith Radar (KAZR)** | 2 s/30 m | Radar reflectivity, Doppler velocity, spectral width |
| **Microwave Radiometer 3-Channel (MWR3C)** | 30 s/column | Brightness temperatures, LWP |
| **Multifilter Rotating Shadowband Radiometer (MFRSR)** | 20 s/column | Narrowband irradiance at 415, 500, 615, 673, 870, and 940 nm, aerosol optical depth, cloud optical depth |
| **Balloon-Borne Sounding System (SONDE)** | 2 times per day | Atmospheric pressure, temperature, and moisture profiles |



**Table 2: The selected 12 flight days and their descriptions**

| Date | Time (UTC) | In-cloud time | Cloud Conditions | Mean Distance between the ENA observatory and G1 |
|---|---|---|---|---|
| **2017/06/21** | 11:34-15:17 | 14 min | Stratocumulus cloud layer | 23.3 km |
| **2017/06/28** | 9:02-12:34 | 10 min | Low-level stratus (broken conditions) | 13.4 km |
| **2017/06/30** | 9:27-13:16 | 1 hour 8 min | Persistent stratus cloud layer with top near 1 km | 13.1 km |
| **2017/07/06** | 8:22-11:58 | 1 hour 1 min | Stratocumulus cloud with embedded drizzle patches | 9.8 km |
| **2017/07/08** | 8:34-12:44 | 37 min | Low-level stratus with cloud top near 1 km (broken conditions) | 147. 7 km |
| **2017/07/18** | 8:31-12:04 | 1 hour 32 min | Drizzling stratocumulus clouds | 14.8 km |
| **2018/01/19** | 12:10-16:06 | 48 min | Drizzling stratocumulus clouds | 3.6 km |
| **2018/01/25** | 11:02-14:49 | 1 hour 25 min | Overcast stratocumulus clouds | 11.9 km |
| **2018/01/26** | 11:05-15:00 | 1 hour 37 min | Overcast stratocumulus clouds | 134.8 km |
| **2018/01/30** | 9:34-13:50 | 1 hour 34 min | Solid stratocumulus cloud deck | 18.5 km |
| **2018/02/07** | 17:28-19:22 | 44 min | Overcast stratocumulus clouds | 17.9 km |
| **2018/02/12** | 11:05-15:07 | 26 min | Low-level stratus (broken conditions) | 2.8 km |



**Table 3: Median $N_d$ values for the selected 12 flight days. The percentages in parentheses represent the relative difference of the $N_d$ retrievals compared to $N_{d\_FCDP}$.**

| Date | $N_{d\_FCDP}$ $cm^{-3}$ | $N_{d\_CAS}$ $cm^{-3}$ | $N_{d\_mpl}$ $cm^{-3}$ | $N_{d\_rl}$ $cm^{-3}$ | $N_{d\_radar}$ $cm^{-3}$ | $N_{d\_vap}$ $cm^{-3}$ |
|---|---|---|---|---|---|---|
| 2017/06/21 | 66 | 75 (13%) | 101 (53%) | n/a | 100 (50%) | 129 (94%) |
| 2017/06/28 | 58 | 63 (8%) | 94 (62%) | 24 (-59%) | 75 (29%) | 44 (-24%) |
| 2017/06/30 | 115 | 125 (9%) | 217 (89%) | 136 (19%) | 95 (-17%) | 314 (174%) |
| 2017/07/06 | 95 | 96 (2%) | 127 (34%) | 106 (12%) | 79 (-17%) | 87 (-8%) |
| 2017/07/08 | 76 | 83 (9%) | 102 (34%) | 16 (-79%) | 101 (32%) | 123 (61%) |
| 2017/07/18 | 67 | 61 (-9%) | 61 (-9%) | 34 (-49%) | 67 (1%) | 62 (-8%) |
| 2018/01/19 | 33 | 50 (52%) | 38 (15%) | 37 (13%) | 67 (102%) | 54 (65%) |
| 2018/01/25 | 57 | 62 (8%) | 33 (-43%) | 52 (-10%) | 64 (12%) | 46 (-20%) |
| 2018/01/26 | 94 | 72 (-23%) | 69 (-27%) | 98 (4%) | 73 (-22%) | 120 (27%) |
| 2018/01/30 | 80 | 74 (-7%) | 54 (-32%) | 60 (-25%) | 75 (-6%) | 57 (-29%) |
| 2018/02/07 | 125 | 72 (-42%) | 76 (-39%) | 133 (7%) | 77(-38%) | 90 (-28%) |
| 2018/02/12 | 105 | 71 (-32%) | 77 (-26%) | 47 (-55%) | 80 (-23%) | 96 (-9%) |




**Figure 1: Cloud properties of the 12 selected flight days derived from the ENA ground-based remote sensing observations during aircraft measurement periods: a) Fractional sky cover obtained from total sky imager (TSI) observations; b) Cloud-base height determined from MPL measurements; c) Cloud depth; d) LWP obtained from the MWRRETv2 VAP; and e) column maximum $Z_e$ ($Z_{e\_max}$) from KAZR measurements. The box-and-whisker plots display the 5th, 25th, 50th, 75th, and 95th percentiles. Dashed lines in a) and e) represent 100% cloud fraction and $Z_{e\_max}$ of 0 dBZ, respectively.**






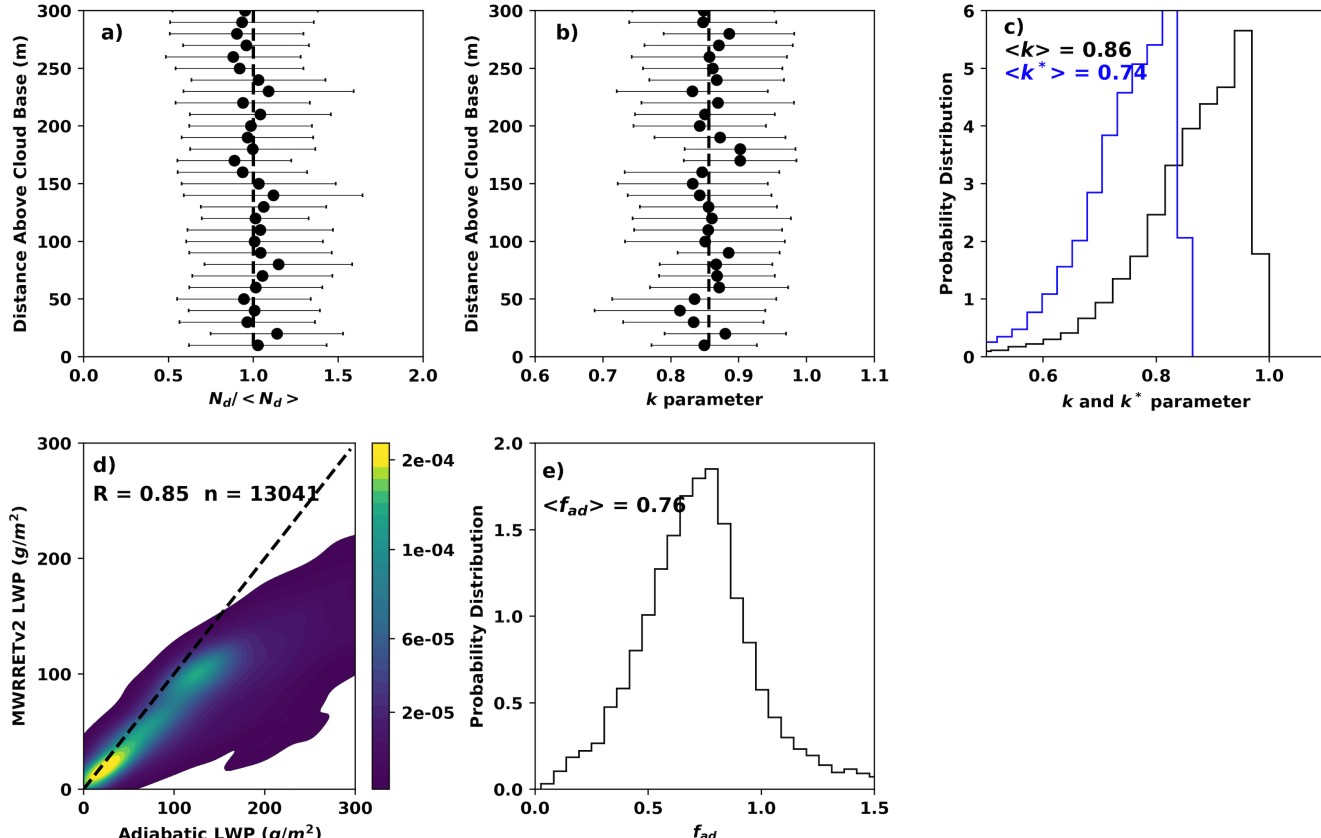

**Figure 2: Statistics of cloud properties used in the retrieval algorithms from *in situ* and ground-based measurements during the 12 selected flight days: a) The mean and standard deviation of the FCDP-measured $N_d$ normalized by the flight average; b) The mean and standard deviation of the derived $k$ parameter profile within clouds; c) The PDFs of the $k$ and $k^*$ parameters; d) The regression between LWPs from MWRRETv2 retrievals and those calculated assuming an adiabatic cloud, where $R$ is the Person correlation coefficient and $n$ the total number of profiles; and e) PDF of $f_{ad}$.**





**Figure 3: An example of ground-based remote sensing measurements and $N_d$ retrievals on January 26, 2018: a) RL extinction coefficient ($\beta_e$) profiles from the RLPROF-FEX VAP; b) MPL $\beta_e$ profiles; c) KAZR radar reflectivity profiles; d) LWPs from MWRRETv2 retrievals ($LWP_{mwr}$) and calculated assuming an adiabatic cloud liquid water**

**content vertical profile ($LWP_{ad}$); e) retrieved $N_{d\_mpl}$, $N_{d\_rl}$, $N_{d\_radar}$, $N_{d\_vap}$; f) derived layer mean $r_e$ ($r_{em}$) per retrieval, $r_{em\_mpl}$, $r_{em\_rl}$, $r_{em\_radar}$, $r_{em\_vap}$. Black lines in a), b), c) are cloud top and base detected with combined lidar and radar measurements. The grey zone indicates the time of concurrent aircraft *in situ* measurements.**



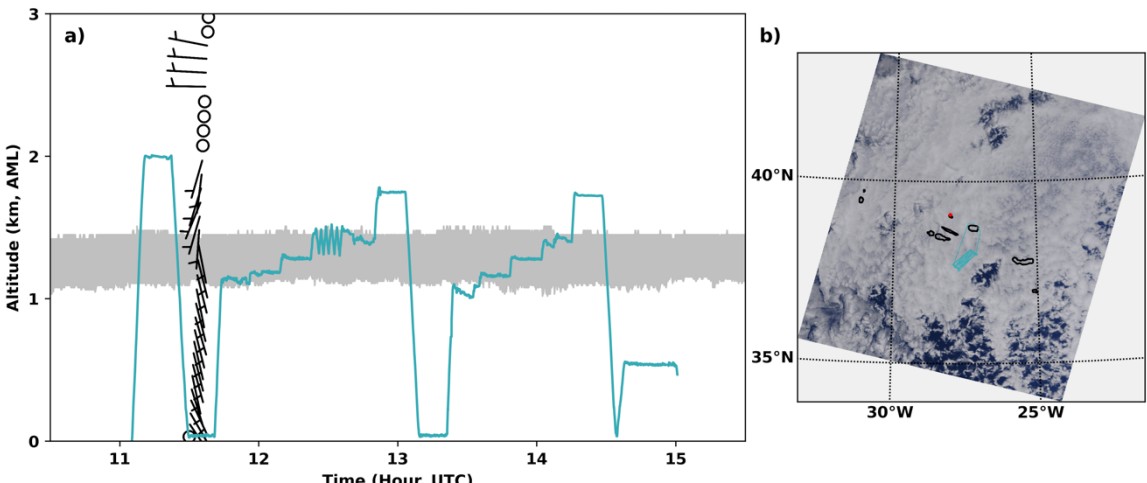

**Figure 4: a) The G-1 aircraft flight track on January 26, 2018. The grey zone represents the cloud layer. Wind barbs are from the ARM radiosonde measurements at 11:30 UTC at the ENA observatory; b) Moderate Resolution Imaging Spectroradiometer (MODIS) true color image of clouds between 13:00-13:10 UTC on January 26, 2018. The red star indicates the location of the ENA observatory. The blue lines in a) and b) represent the aircraft flight track.**



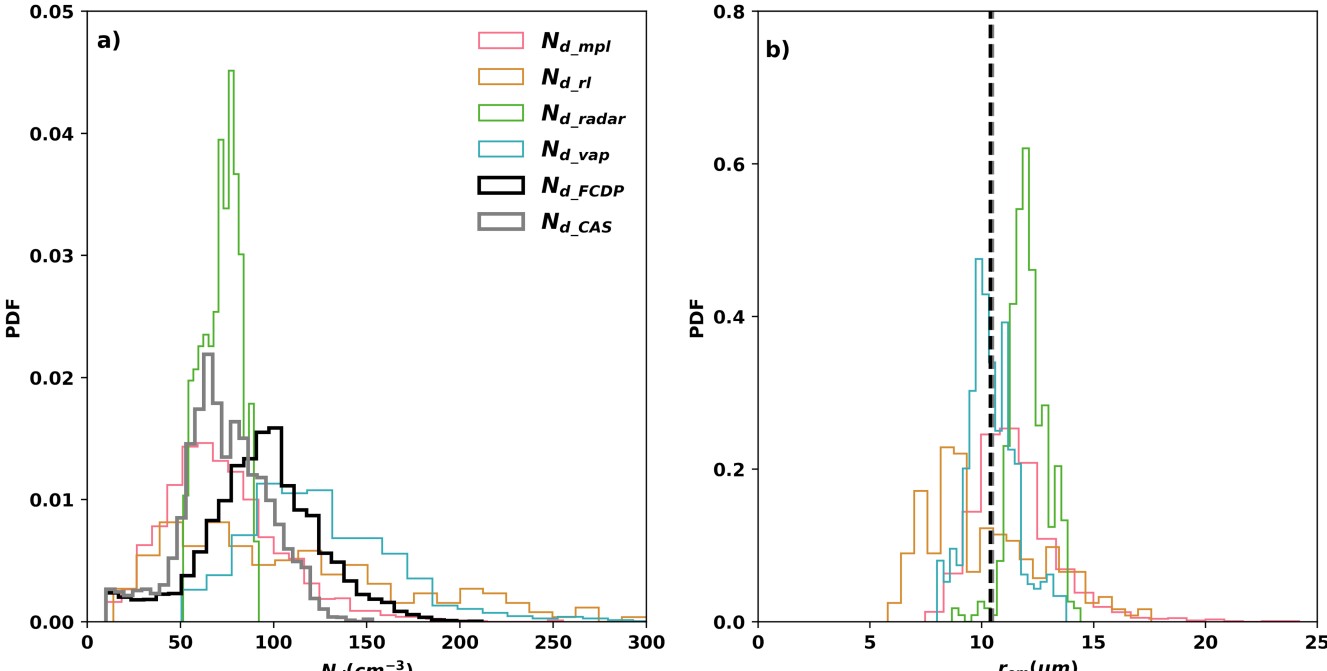

**Figure 5: Evaluation of ground-based retrievals of $N_d$ and $r_e$ with aircraft *in situ* measurements for the case on January 26, 2018. a) PDFs of $N_{d\_mpl}$, $N_{d\_rl}$, $N_{d\_radar}$, $N_{d\_vap}$, $N_{d\_FCDP}$, and $N_{d\_CAS}$ during the time of concurrent aircraft measurements; b) PDFs of $r_{em\_mpl}$, $r_{em\_rl}$, $r_{em\_radar}$, $r_{em\_vap}$, and derived $r_{em\_FCDP}$ and $r_{em\_CAS}$ from *in situ* probe measurements. The colors of $r_{em}$ lines in b) correspond to those given in a). Dashed lines in b) are mean $r_e$ from FCDP and CAS measurements during all in-cloud penetrations.**







**Figure 6: Evaluation of ground-based retrievals of $N_d$ and $r_e$ with aircraft *in situ* measurements for the 12 selected flight days. a) Boxplots of $N_{d\_mpl}$, $N_{d\_rl}$, $N_{d\_radar}$, $N_{d\_vap}$, $N_{d\_FCDP}$, and $N_{d\_CAS}$ during the time of concurrent aircraft measurements; b) Boxplots of $r_{em\_mpl}$, $r_{em\_rl}$, $r_{em\_radar}$, $r_{e\_vap}$, and derived $r_{em\_FCDP}$ and $r_{em\_CAS}$ from *in situ* probe measurements. The colors of $r_{em}$ boxplots in b) correspond to those given in a).**







**Figure 7: Monthly variations of overcast marine boundary layer cloud $N_d$ and $r_e$ using four years of ground-based measurements at the ENA observatory. a) Occurrence of MBL clouds; b) retrieved $N_{d\_mpl}$, $N_{d\_rl}$, $N_{d\_radar}$, and $N_{d\_vap}$; c) cloud condensation nuclei ($N_{ccn}$) from the ARM CCN counter (CCN-100) of aerosol observing system (AOS) at a supersaturation of 0.1%; d) derived $r_{em\_mpl}$, $r_{em\_rl}$, $r_{em\_radar}$, and $r_{em\_vap}$; and e) $Z_{e\_max}$.**