# Peer review of "Evaluation of Four Ground-based Retrievals of Cloud Droplet Number Concentration in Marine Stratocumulus with Aircraft *In Situ* Measurements"

_EGUsphere, 2023_

## Author Comment (AC1)

**Reply to Reviewer #1's comments**

*Liquid water cloud plays an important role in the Earth's atmosphere, while a great deal of uncertainty still exists in observational cloud properties. Cloud droplet number concentration ($N_d$) is one of the most important cloud properties, which associate clouds with aerosol. This study compared four ground-based $N_d$ retrievals from both lidar and radar retrievals with in situ measurements and investigate seasonal variations of $N_d$ and $r_e$. Their results showed good agreement between ground-based retrievals and in situ measurement for overcast conditions. Also, the consistency between $N_d$ retrievals and in situ measurement struggles with broken or low LWP clouds. By extending these retrievals to longer time period, obvious seasonal variations of $N_d$ ($r_e$) values exhibits and are consistent with previous researches. I believe their evaluation promote our understanding of uncertainties of remote sensing data. However, the paper needs to be improved to be qualified for publication by addressing the following comments.*

**We thank the reviewer for these constructive suggestions and comments. We carefully revised the manuscript according to the reviewer's comments.**

*General comments:*

1.  *Line 93-94: I think you need add more details about why you choose these four ground-based $N_d$*

    **Response: We appreciate the reviewer's suggestion. Our analysis encompasses four major ground-based $N_d$ retrievals. We have updated the statement in line 100 to reflect this. Furthermore, in line 98-99, we emphasized the significance of these retrievals with the sentence of 'considering their potential for operational applications and ease of use across different locations '. However, we did not include lidar-based $N_d$ retrievals that either utilize dual-field-of-view lidar extinction profiles or rely on depolarization measurements from lidar multiple scattering. This is due to the specific requirement of the dual-field-of-view lidar configuration and the substantial calibration efforts needed for lidar depolarization measurements. We've incorporated this clarification into the manuscript in line 102-108.**

2.  *Line 121: literatures or documents of the instruments' information showed in Table1should be cited here.*

    **Response: We added references to these instrument handbooks.**

3.  *Line 207-210: This sentence is not easy to read. You may consider reorganizing the sentence structure to simplify and make it clearer.*

    **Response: We reorganized the sentence structure to make it clearer.**

4.  *Line 214: you assume a linear increase of LWC in radar retrievals. Are there any impacts of this assumption to the results without regard to $f_{ad}$ in this situation?*

**Response: We appreciate the reviewer's comment. Given that both LWC and $\sqrt{Z}$ are influenced by $f_{ad}$ in a consistent manner, $f_{ad}$ doesn't affect radar-based $N_d$ retrievals.**

5. *Line 218: you missed $\rho_w$ in equation 7 according to Mace (2000).*

   **Response: We added $\rho_w$ in equation 7.**

6. *Line 232-233: I think you should explain more about the meaning of k* and point out why use k* to replace k.*

   **Response: We added a sentence to explain the meaning of $k^*$ in line 263:**

   **'$k^*$ is the cloud system $k$ parameter, which is the cube of the ratio between the layer-mean volume radius and the layer-mean effective radius.'**

   **In line 264, we referenced Brenguier et al. (2011), noting, 'As both $\tau$ and LWP represent vertical integrals through the entire cloud layer, Brenguier et al. (2011) propose using the cloud system $k^*$ parameter in place of $k$ in equation (8)'. Consequently, the NDROP VAP retrievals utilize the cloud system $k^*$ parameter, while other methods deploy the local mean $k$ parameter. We added this sentence in line 265-266.**

7. *What do the black circles mean in figure 4b?*

   **Response: Black circles represent islands in the region in Figure 4b. We added this clarification in Figure 4b's caption.**

8. *Line 365: the word "greatest" may cause misunderstanding. You should replace it with another word.*

   **Response: We rewrote the sentence as: '$N_{d\_vap}$ retrieval exhibits the highest values'.**

9. *Line 380: I notice that the higher $N_d$ from in situ measurements actually appear on 02/07/2018, 06/30/2017 and 02/12/2018. If you have a specific criterion, you should point out here.*

   **Response: We revised the sentence as following:**

   **'with generally higher $N_d$ observed on summer IOP days, and lower $N_d$ on winter IOP days'**

10. *For more intuitive and easy reading, I think you should label the broken conditions in Table 3 and other figures that appears the date of 12 flight days.*

    **Response: We appreciate the reviewer's suggestion. We added a * to label broken conditions in Table 2 and 3, and Figure 6.**

11. *Line 412-414: what are the possible causes of the inconsistency of $r_{em}$ and $N_d$ retrievals of NDROP VAP compared to FCDP?*

   **Response: We thank the reviewer for raising the question. We realize that the values of $r_{e\_vap}$ are also slightly greater than those of $r_{e\_FCDP}$ in general. This is primarily because $r_{e\_vap}$ is calculated from measured LWP and $\tau$, both of which are more heavily influenced by the cloud's upper regions where larger droplet particles are prevalent. We revised the sentence in the manuscript.**

*Detail comments:*

1. *Line 28: delete the repeated "using the".*

   **Response: We deleted the repeated words as suggested.**

2. *Line28-30: this sentence has a linguistic flaw. I suppose you may want to begin a new sentence from "given".*

   **Response: We changed the sentence structure by starting a new sentence for the reasons why we recommend the Micropulse lidar-based method.**

3. *Line 59: cloud optical -> cloud optical depth*

   **Response: Thanks for pointing out the typo. We changed 'cloud optical' to 'cloud optical depth' in the text.**

4. *Line 95: 2018 -> 2017*

   **Response: We changed it from '2018' to '2017 as suggested.**

5. *Line 219: Miles -> Mace*

   **Response: Our equation was sourced directly from Miles et al. (2000). We realized we omitted the reference to Miles et al. (2000) in our initial reference list and have now included it in this revision.**

6. *Line 289: missing 'cloud depth' in your statement of figure 1.*

   **Response: We added 'cloud depth' in that sentence.**

7. *Line 293: figure 1c -> figure 1d*

   **Response: We changed 'figure 1c' to 'figure 1d'.**

8. *Line 421: full name of TSI should be presented in your main body.*

   **Response: We added the full name of TSI in the sentence.**

---

## Author Comment (AC2)

*Reply to Reviewer #2's comments*

*This study aims to compare and evaluate four ground-based remote-sensing methods for retrieving cloud properties, with a focus on CDNC retrievals. CDNC is crucial for studying aerosol-cloud interactions and for understanding cloud processes but its retrieval from remote sensing still suffers from significant uncertainties. Numerous methods exist for CDNC retrieval that rely on a number of assumptions often unclear to the community. Therefore, this effort to summarize and evaluate here these methods against in-situ observations from 12 flights is timely and valuable.*

*The manuscript is well-written and appropriately cites current literature. The authors carefully describe the four established retrieval techniques and their respective assumptions. Comparisons to in-situ observations are done meticulously. I find the study convincing and well within the scope of AMT. I advise for publication after addressing the following comments and suggestions.*

**We thank the reviewer for constructive suggestions and comments. We carefully revised the manuscript according to the reviewer's comments.**

*General comments:*

1. *l. 146-147: This sentence is somewhat misleading as it could imply that this formula is independent of the DSD shape. While the equation may not require assumptions about the DSD shape, these will be needed when retrieving or computing the extinction coefficient and liquid water content. If that is what the author meant I would suggest rephrasing for clarity. If not, please provide more explanations.*

   **Response: We concur with the reviewer's observation regarding the necessity of DSD shape assumptions when computing extinction and liquid water content, as evident in equations (2) and (3). Nevertheless, equation (6), which calculates $N_d$, introduces the *k* parameter, thereby eliminating the dependence on a pre-assumed DSD shape.**

2. *l. 216: Does the logarithmic width of the lognormal distribution (0.38) relate to the k parameter? It would be interesting to know its equivalent value if so.*

   **Response: We are grateful to the reviewer for providing the constructive suggestion. By assuming a lognormal droplet size distribution, the relationship between $\sigma_x$ and the *k* parameter can be expressed as:**

$$k = exp(-3 * \sigma_x^2)$$

   **A value of 0.38 for $\sigma_x$ is equivalent to a *k* value of 0.65. Martin et al. (1994) showed that *k* ranges from 0.67 ± 0.07 in continental air masses to 0.80 ± 0.07 in the marine ones. Consequently, we updated the radar-based $N_d$ dataset using $\sigma_x$ of 0.23**

(corresponding to a *k* value of 0.86 under a lognormal DSD condition) as discussion in lines between 247 and 251 in this revision. Unfortunately, we are not able to eliminate the need for pre-assuming a DSD shape for the radar-based $N_d$ retrievals using the *k* parameter like that for the lidar-based retrievals.

Reference:

Martin, G. M., D. W. Johnson, and A. Spice, 1994: The Measurement and Parameterization of Effective Radius of Droplets in Warm Stratocumulus Clouds. *J. Atmos. Sci.*, 51, 1823–1842, https://doi.org/10.1175/1520-0469(1994)051<1823:TMAPOE>2.0.CO;2.

3. *l. 233: Briefly explain the physical meaning of the cloud system k parameter, as defined by Brenguier et al., and why it's used for the VAP retrievals but not the other approaches.*

   Response: The *k* parameter is an empirical correction factor to account for the changes of cloud droplet spectrum. As mentioned in line 158, 'a parameter *k* is introduced to link $\beta$ and $LWC$ that is a measure of the width of the cloud DSD'. Based on the formula provided in response to question 2, *k* is inversely proportional to the width of the DSD spectra.

   We've also clarified the significance of *k\** in line 263, stating:

   '$k^*$ is the cloud system *k* parameter, which is the cube of the ratio between the layer-mean volume radius and the layer-mean effective radius.'

   In line 264, we referenced Brenguier et al. (2011), noting, 'As both $\tau$ and LWP represent vertical integrals through the entire cloud layer, Brenguier et al. (2011) propose using the cloud system *k\** parameter in place of *k* in equation (8)'. Consequently, the NDROP VAP retrievals utilize the cloud system *k\** parameter, while other methods deploy the local mean *k* parameter.

4. *l. 240-245: Specify the field of view for the instruments or the spatial resolution of the COD and LWP retrievals. It would also be useful to briefly state what the basic assumptions are for these retrievals, especially regarding the vertical distribution of cloud properties: are clouds assumed to be vertical homogeneous? In that case there would be an inconsistency with the assumptions from Eq 8, where the COD and LWP are used. This should at least be mentioned, as it can also partly explain why the VAP Nd retrievals are often different from (and more uncertain than) others.*

   Response: Between lines 275 and 283, we included the field-of-view specifications for the instruments and clarified the assumptions underpinning the COD and LWP retrievals. Notably, neither of these retrievals employ the assumption of vertical homogeneity within the clouds.

5. *l. 256-263: Opting for 9/5 over 3/2 would align better with the adiabatic assumption in Eq. 8. You mention Chiu et al. (2012) found better results using 9/5 but still chose to use 3/2. Please justify this choice.*

   **Response: We are grateful for the reviewer's observation. Upon revisiting the equations in Wood and Hartmann (2006), we discerned that the '9/5' factor is applied to $r_{e,top}$, derived from the MODIS product, where $r_{e,top}$ signifies the 'near-cloud-top effective radius.' Our objective is to compare the layer-mean $r_e$ across lidar-based, radar-based, and VAP retrievals, thus we continue to employ the '3/2' factor to maintain consistency among all retrievals. The sentences pertaining to the '9/5' factor have been removed in the revision.**

   **Reference:**

   **Wood, R., and D. L. Hartmann, 2006: Spatial Variability of Liquid Water Path in Marine Low Cloud: The Importance of Mesoscale Cellular Convection. J. Climate, 19, 1748–1764, https://doi.org/10.1175/JCLI3702.1.**

6. *l. 280: Indicate if clouds were fully profiled vertically, to clarify the later definition of <Nd>.*

   **Response: We reviewed the flight tracks for all 12 flight days and confirmed that each day had at least one complete traversal from the cloud base to the cloud top. We have incorporated this observation into the manuscript at line 332.**

7. *l. 317-324: It's unclear whether values of fad > 1 were set to 1 before computing the mean of 0.76. This can be problematic because the mean value may then not be very meaningful (the distribution would be far from normal). Why not use the slope from Fig 2d instead?*

   **Response: We calculated the mean $f_{ad}$ without constraining $f_{ad}$ to 1 in instances where $f_{ad}$ > 1. We're grateful for the reviewer's suggestion to utilize the slope from Fig 2d. The linear fit between MWRRET LWP and adiabatic LWP yields a modest slope of 0.47 and an intercept of 19.6. Our intention with Figure 2d is to underscore the strong correlation between WMRRET LWP and adiabatic LWP. To this end, we've included the statement in line 368:**

   **'The LWP from WMRRET and the adiabatic LWP show a strong correlation, evidenced by a Pearson correlation coefficient of 0.85'.**

   **Additionally, as noted in line 366, "we determine $f_{ad}$ using the ratio of the instantaneous MWRRETv2 LWP to the computed adiabatic LWP." Thus, neither the mean $f_{ad}$ nor the slope affects the $N_d$ retrievals.**

8. *l. 384: Are these truly seasonal variations, or are rather day-to-day as mentioned later?*

**Response: Owing to the scarcity of *in situ* observations, such as the 12 flight days in our study and the 39 flight days in Wang et al. (2022), we observe day-to-day variations. Nevertheless, when comparing the two IOP durations, a trend emerges: $N_d$ during the summer IOP tends to be higher than on the winter IOP. This pattern is further corroborated when we apply the retrievals to four years of ENA data, as depicted in Figure 7b.**

**Reference:**

**Wang, J., et al.: Aerosol and Cloud Experiments in the Eastern North Atlantic (ACE-ENA), *Bulletin of the American Meteorological Society, 103*(2), E619-E641, doi: https://doi.org/10.1175/BAMS-D-19-0220.1, 2022.**

9. *l. 501-503: Do you think the issues faced by VAP method are comparable to those faced by typical radiometer-based satellite retrievals of Nd? This conclusion may not be straightforward to draw but could be an very interesting message for the community. As you note in the next paragraph, ground-based retrievals are very valuable to evaluate the global satellite dataset.*

   **Response: We believe the issues faced by the VAP method (e.g., subpixel heterogeneity, and viewing geometry) are not as pronounced as those in the typical radiometer-based satellite retrievals of $N_d$, as highlighted by Grosvenor et al., (2018). This is because MFRSR and MWR measurements used in the VAP method have fixed viewing geometry. On the other hand, satellite retrievals are subject to more significant challenges arising from viewing geometry and subpixel heterogeneity. Furthermore, their accuracy may be compromised by potential biases introduced by upper-level layers of thin cloud and aerosol. Nevertheless, it is noted that satellite retrievals do not mix hemispheric and nadir measurements, in contrast to the VAP approach. This characteristic results in fewer complications related to varying instrument fields of view (FOVs). Given that our study does not focus on evaluating satellite retrievals of $N_d$, we did not include this discussion in our manuscript.**

   **Reference:**

   **Grosvenor, D. P., Sourdeval, O., Zuidema, P., Ackerman, A., Alexandrov, M. D., Bennartz, R., et al. (2018). Remote sensing of droplet number concentration in warm clouds: A review of the current state of knowledge and perspectives. Reviews of Geophysics, 56, 409–453. https://doi.org/10.1029/2017RG000593**